# Adverse Reaction Profiles Related to Gastrointestinal Bleeding Events Associated with BCR-ABL Tyrosine Kinase Inhibitors

**DOI:** 10.3390/medicina58101495

**Published:** 2022-10-20

**Authors:** Kei Kawada, Tomoaki Ishida, Kohei Jobu, Shumpei Morisawa, Naohisa Tamura, Shouhei Sugimoto, Masafumi Okazaki, Saburo Yoshioka, Mitsuhiko Miyamura

**Affiliations:** 1Graduate School of Integrated Arts and Sciences, Kochi University, Kohasu, Oko Town, Nankoku City 783-8505, Japan; 2Department of Pharmacy, Kochi Medical School Hospital, 185-1 Kohasu, Oko Town, Nankoku City 783-8505, Japan

**Keywords:** gastrointestinal bleeding, BCR-ABL tyrosine kinase inhibitor, dasatinib, imatinib, Japanese Adverse Drug Event Report database, chronic myeloid leukemia

## Abstract

*Background and Objectives*: The aim of this study is to investigate the characteristics of gastrointestinal bleeding events associated with BCR-ABL tyrosine kinase inhibitor (TKI) treatment, using the reporting odds ratio (ROR) of the adverse event reports submitted to the Japanese Adverse Drug Event Report database between 2004 and 2020, and to examine the number of reported TKI-related gastrointestinal bleeding cases according to sex and age, as well as the actual number of TKI prescriptions issued in Japan. *Materials and Methods*: The RORs and 95% confidence intervals (CIs) of gastrointestinal bleeding events related to TKIs were calculated using the data of the 595,121 included cases. *Results*: Significant gastrointestinal bleeding events were detected for dasatinib (crude ROR: 4.47, 95% CI: 3.77–5.28) and imatinib (crude ROR: 1.22, 95% CI: 1.01–1.46). In multiple logistic regression analyses, significant gastrointestinal bleeding events were detected for dasatinib (adjusted ROR: 8.02, 95% CI: 5.75–10.2), imatinib (adjusted ROR: 1.81, 95% CI: 1.2–2.72), age (≥60 years, adjusted ROR: 2.22, 95% CI: 2.1–2.36), reporting year (adjusted ROR: 1.04, 95% CI: 1.04–1.05), and male sex (adjusted ROR: 1.47, 95% CI: 1.37–1.57). Interaction analysis revealed that the association of gastrointestinal bleeding with dasatinib was affected by age (≥60 years) and sex (female), with the number and proportion of dasatinib-related gastrointestinal bleeding cases increasing among those aged ≥60 years. *Conclusions*: Specific TKIs and patient characteristics were associated with gastrointestinal bleeding. Our results aid the prompt identification and treatment of TKI-related gastrointestinal bleeding.

## 1. Introduction

Chronic myeloid leukemia (CML) is a clonal disease characterized by the presence of the Philadelphia (Ph+) chromosome and its oncogenic product, BCR-ABL. BCR-ABL is a constitutively active tyrosine kinase present in >90% of patients with CML. Treatment of CML was revolutionized by the introduction of a BCR-ABL tyrosine kinase inhibitor (TKI). The clinical use of TKIs has resulted in notably improved prognosis, response rate, overall survival, and patient outcomes in patients with CML. However, TKI use is associated with various adverse effects (AEs), including hematological and non-hematological AEs (i.e., gastrointestinal disturbances, cardiotoxicity, liver toxicity, pleural effusion, edema, and muscle cramps); and the monitoring and management of these AEs are crucial for the success of CML treatment [1]. In particular, TKI-related gastrointestinal bleeding has been reported in severe and fatal cases and therefore requires careful management [2,3]. However, there are only a few previous studies on gastrointestinal bleeding relating to TKI use. Previous studies included small sample sizes [2,4,5], and as gastrointestinal bleeding occurred infrequently in patients enrolled in those studies, sufficient data are lacking. Furthermore, in a meta-analysis study, only the ratio of TKI-related gastrointestinal bleeding was reported, and characteristics such as patient background have not been sufficiently investigated [6].

We used data from the Japan Adverse Drug Report (JADER) database to increase the sample size in this study. JADER is a database collecting information on adverse drug reaction cases reported voluntarily to the Pharmaceuticals and Medical Devices Agency (PMDA). Some previous studies have used JADER to investigate rare AEs [7,8,9,10]. Accordingly, using JADER data, it is possible to investigate the relationship between gastrointestinal bleeding and TKIs, as well as the relationship between this AE and patient background characteristics.

The purpose of the current study is to evaluate the relationship between TKIs and gastrointestinal bleeding using the reporting odds ratio (ROR), and to analyze the time-to-onset of hemorrhagic gastrointestinal bleeding. Furthermore, the number of prescription drugs, using data from the 5th National Database of Health Insurance Claims and Specific Health Checkups of Open Data from 2018 (https://www.mhlw.go.jp/ndb/opendatasite, accessed on 31 December 2021), and the actual number of TKI prescriptions in Japan were examined. The number of reported gastrointestinal bleeding cases related to TKIs according to sex and age was also investigated.

## 2. Materials and Methods

### 2.1. Data Source

Among the 667,729 cases in the JADER dataset, 72,608 were excluded because of missing data on sex or age. Similar to the analysis performed in previous studies [8,10,11], the RORs and 95% confidence intervals (CIs) of gastrointestinal bleeding events related to TKIs were calculated using data from the 595,121 included cases. For the analysis of the onset of drug-induced gastrointestinal bleeding, four TKIs (dasatinib, imatinib, bosutinib, and nilotinib) were selected from the JADER dataset.

First, to evaluate the effect of age and sex on gastrointestinal bleeding, reports were stratified into the following age and sex groups: groups 0–59 years and ≥60 years, as well as male and female groups. We then identified the TKIs significantly associated with gastrointestinal bleeding. Second, whether specific TKIs, age ≥60 years, and male sex were significantly associated with gastrointestinal bleeding was examined using multivariate analysis with adjusted RORs. Subsequently, the interaction of possible factors associated with gastrointestinal bleeding was examined. Third, the onset time of gastrointestinal bleeding associated with TKIs was investigated using Weibull distribution analysis. Finally, the number of TKI prescriptions and the outcome of TKI-associated gastrointestinal bleeding were investigated.

We used the ICH Medical Dictionary for Regulatory Activities (MedDRA) v24.0 to extract the AEs and underlying diseases listed in the JADER database. Gastrointestinal bleeding was defined by preferred terms in the Standardized MedDRA Queries (SMQ) suggestive of gastrointestinal hemorrhage (SMQ: 20000108).

### 2.2. Data Extraction

The JADER database used in this study collects adverse drug reaction data reported voluntarily to the PMDA. Data recorded from April 2004 to March 2020 in the JADER database were downloaded from the PMDA website (http://www.pmda.go.jp/) (accessed on 1 December 2021). The JADER dataset consists of four tables containing the following data: (1) patient information, including sex, age, and body weight; (2) patient drug information; (3) AEs and outcomes; (4) medical history and primary illness. These four tables were integrated using the FUND E-Z Backup Archive (FUND E-Z Development Corporation, White Plains, NY, USA).

### 2.3. Analysis of the Reporting Odds Ratio

For the analysis of gastrointestinal bleeding associated with TKI administration, the crude RORs and 95% CIs were calculated. To calculate the crude ROR, first, the cases were classified into groups (a) to (d) as follows: (a) individuals who received the drug of interest and exhibited the AE of interest; (b) individuals who received the drug of interest and exhibited other AEs (of no interest); (c) individuals who received other drugs (of no interest) and exhibited the AE of interest; and (d) individuals who received other drugs (of no interest) and exhibited other AEs (of no interest). Next, the crude ROR was calculated using the following equation [12]:Crude ROR = (a/b)/(c/d)
95% CI = exp [log (ROR) ± 1.96 √((1/a) + (1/b) + (1/c) + (1/d))]

The RORs were expressed as point estimates with 95% CIs. The data were analyzed using Fisher’s exact test.

### 2.4. Analysis of the Adjusted Reporting Odds Ratio

We calculated the adjusted ROR based on a previous report [7]. In addition, the patients were stratified by age into 0–59 and ≥60 years groups [9]. To construct the logistic model, sex (male), reporting year [10], and stratified age groups were coded. The following logistic model was used for the analysis:Log (odds) = β0 + β1Y + β2S + β3A
where Y = reporting year, S = sex, and A = stratified age group.

The 0–59-year group was used as a reference to calculate the RORs adjusted for age variations, and the sex = female group was used as a reference to calculate the RORs adjusted for sex variations. Subsequently, the interaction of factors for gastrointestinal bleeding was examined; results with *p* < 0.05 were considered statistically significant.

### 2.5. Analysis of Onset Time of TKI-Associated Gastrointestinal Bleeding

The gastrointestinal bleeding onset time was defined as the period from the date of the first TKI prescription to the date of occurrence of gastrointestinal bleeding. Cases with complete data on age, sex, TKI use (suspected as the cause of the gastrointestinal bleeding event), complete information on gastrointestinal bleeding occurrence, and prescription start date were included in the time-to-onset analysis. It is important to choose the optimal length of the evaluation period in order to evaluate the time-to-onset of gastrointestinal bleeding. We chose an analysis period of 365 days after administration, based on a previous study [9]. Median duration, interquartile range (IQR), and Weibull shape parameters (WSPs) were used to evaluate the onset data. The scale parameter α of the Weibull distribution determines the scale of the distribution function. A larger α value stretches the distribution, whereas a smaller α value shrinks the data distribution. The WSP β parameter of the Weibull distribution determines the shape of the distribution function. Larger and smaller shape values produce left- and right-skewed curves, respectively. The shape parameter β of the Weibull distribution was used to indicate the level of hazard over time without a reference population. When β is equal to 1, the hazard is estimated to be constant over time. If β is greater than 1, and the 95% CI of β excludes the value 1, the hazard is considered to increase with time [13].

### 2.6. Analysis of Age and Outcomes in Cases with Dasatinib-Associated Gastrointestinal Bleeding

Cases with complete data on age, dasatinib use, and gastrointestinal bleeding occurrence were used in this analysis (160 cases). Cases were classified according to age and outcome. Outcomes recorded in the table of adverse reactions in the JADER database, such as “recovered”, “remission”, “not recovered”, “sequela”, “death”, and “unclear”, were used.

### 2.7. Statistical Analyses

The results were considered statistically significant at *p* < 0.05. The analyses were performed using JMP 14.0 (SAS Institute, Cary, NC, USA). A signal was detected when the lower limit of the 95% CI of the adjusted ROR exceeded 1.

## 3. Results

### 3.1. Identification of Possible Factors for Gastrointestinal Bleeding

Of the 595,121 cases reported between April 2004 and December 2020, 15,418 exhibited gastrointestinal bleeding. The reporting rates of gastrointestinal bleeding related to dasatinib, imatinib, bosutinib, and nilotinib were 10.55, 3.13, 3.59, and 1.70%, respectively (Table 1). Significant gastrointestinal bleeding signals were detected among patients receiving dasatinib (crude ROR: 4.47, 95% CI: 3.77–5.28, *p <* 0.001) and imatinib (crude ROR: 1.22, 95% CI: 1.01–1.46, *p =* 0.038). The crude RORs (95% CI) for dasatinib in the 0–59-year-old group and ≥60-year-old group were 6.30 (4.72–8.28) and 4.07 (3.27–5.02), respectively. The crude RORs (95% CI) for dasatinib in the male group and the female group were 3.43 (2.70–4.33) and 6.15 (4.78–7.81), respectively. The crude ROR (95% CI) for imatinib in the 0–59-year-old group was 1.66 (1.15–2.33), and that in the female group was 1.46 (1.09–1.91; Table 1).

### 3.2. Analysis of the Adjusted Reporting Odds Ratio for Gastrointestinal Bleeding

In multivariate logistic regression analysis, significant contributions were observed for dasatinib (adjusted ROR: 8.02, 95% CI: 5.75–10.2, *p <* 0.001), imatinib (adjusted ROR: 1.81, 95% CI: 1.2–2.72, *p =* 0.0045), age (≥60 years, adjusted ROR: 2.22, 95% CI: 2.1–2.36, *p <* 0.001), reporting year (adjusted ROR: 1.04, 95% CI: 1.04–1.05, *p <* 0.001), and sex (male, adjusted ROR: 1.47, 95% CI: 1.37–1.57, *p <* 0.001; Table 2). The interaction results between age (≤60 years) and sex (male) (*p <* 0.0001), age (≥60 years) and dasatinib (*p =* 0.035), and sex (male) and dasatinib (*p <* 0.0001) were also significant.

### 3.3. Analysis of the Onset Time of Dasatinib- and Imatinib-Associated Gastrointestinal Bleeding

The time-to-onset profiles are depicted in Figure 1. The onset of drug-induced gastrointestinal bleeding was analyzed using Weibull distribution analysis. The median and quartile of gastrointestinal bleeding were 81.0 (23.0–141.0) days after dasatinib use and 33.0 (13.0–136.0) days after imatinib use. The onset of gastrointestinal bleeding occurred in 66% (64/97) of cases, 30 days post-dasatinib administration, and in 52% (16/31) of cases, 30 days post-imatinib administration.

### 3.4. Analysis of Age and Outcomes in Cases with Dasatinib-Associated Gastrointestinal Bleeding

The outcome profiles of cases with dasatinib-associated gastrointestinal bleeding are summarized according to sex, age, and outcome in Figure 2. Although the outcome “sequela” was listed in the JADER database, there were no cases of dasatinib-associated gastrointestinal bleeding in which the outcome corresponded to “sequela.” The group aged >60 years showed a significantly higher number of dasatinib-associated gastrointestinal bleeding cases and a greater proportion of severe cases than the group aged <60 years.

## 4. Discussion

In this study, the association between TKIs and gastrointestinal bleeding was evaluated using data from the JADER database. The results of the analysis of RORs suggested that dasatinib and imatinib increase the incidence of gastrointestinal bleeding events. In particular, the association of gastrointestinal bleeding with dasatinib was affected by sex and age, and gastrointestinal bleeding became more frequent in patients aged ≥60 years and in females.

This study showed signals associated with gastrointestinal bleeding in patients receiving dasatinib and imatinib among TKIs. In a previous study, these TKIs were reported to be associated with gastrointestinal bleeding, and it was suggested that gastrointestinal or other bleeding events occurred in 5–23% of patients in the median age range between 46–63 years, with an equal male-to-female ratio [2,4,14]. However, the relationship between gastrointestinal bleeding and patient background was not clear because of the limited number of cases in previous studies. Using the JADER database to increase the sample size, we found that the reported rate of gastrointestinal bleeding with imatinib and dasatinib was 3.13% and 10.55%, respectively, and the association of gastrointestinal bleeding with dasatinib was affected by sex and age (≥60 years).

We also applied time-to-onset analysis to validate the results. Our results provide novel insights into the time-to-onset of gastrointestinal bleeding, and can be summarized as follows: gastrointestinal bleeding associated with dasatinib and imatinib developed early post-administration and decreased gradually thereafter; however, more than half of the gastrointestinal bleeding cases were reported 30 days after the first drug administration in the real-world dataset. Gastrointestinal bleeding associated with dasatinib and imatinib occurred later than that associated with anticoagulants (12–47.5 days) [8], suggesting that patients receiving dasatinib and imatinib should be monitored for long-term bleeding complications, specifically because dasatinib and imatinib are often prescribed for outpatients.

The pathophysiology of bleeding associated with dasatinib and imatinib therapy remains poorly understood. However, previous in vitro and ex vivo studies have demonstrated that dasatinib treatment resulted in the dose-dependent impairment of platelet activation and aggregation [3]. Furthermore, dasatinib, in contrast to other TKIs, is also known to disrupt the barrier function of the vascular endothelium [15]. In addition, imatinib induced platelet aggregation abnormalities as per platelet aggregometry testing and caused platelet dysfunction [16,17]. These effects of dasatinib and imatinib might contribute to an increase in the incidence of gastrointestinal bleeding. Furthermore, differences in platelet dysfunction caused by dasatinib or imatinib have been reported [18]. These differences may be related to the difference in the time-to-onset of gastrointestinal bleeding between dasatinib and imatinib.

There were differences in the number of reported cases and the time-to-onset of gastrointestinal bleeding between dasatinib and imatinib in this study. In a previous study, dasatinib affected platelet function through the inhibition of the SRC family of kinases (which play a crucial role in supporting fibrin clot retraction) and induced hemostatic defects. Conversely, imatinib did not act on the SRC family of kinases. Furthermore, patients treated with imatinib showed a reduction in thrombus growth compared with that in healthy donors, but the effect of dasatinib was much more pronounced in this aspect [1]. The differences in the effects of dasatinib and imatinib on platelet function and thrombus growth may affect the number of reported cases and the time to onset of gastrointestinal bleeding.

Our analysis of the RORs after adjusting for the number of gastrointestinal bleeding cases provided information on the association between gastrointestinal bleeding and sex. Adjusting the ROR for the number of reported gastrointestinal bleeding cases revealed that sex (female) can influence gastrointestinal bleeding. Dasatinib is mainly metabolized by the cytochrome P450 enzyme CYP3A4 to active metabolites in the liver [19], and interestingly, CYP3A4 levels in men are reported to be 50% of those in women [20,21]. This may suggest a dissimilarity in the ability to metabolize dasatinib due to sex differences, which may affect the risk of gastrointestinal bleeding.

The number of gastrointestinal bleeding cases associated with dasatinib were highest among patients aged ≥60 years, despite being prescribed to those under 60 years of age (Appendix A). In a previous study, older patients receiving dasatinib were reported to show a higher incidence of AEs than younger patients [22,23]. In addition, low-dose (e.g., ≤20 mg/day) dasatinib therapy generated an adequate molecular response in most older patients with chronic phase CML without causing AEs [24,25]. These results suggest that the dose of dasatinib for older adults may need to be selected carefully because older patients have a reduced ability to metabolize drugs [26].

There were several limitations to this study. First, there was no evidence that the reported event was caused by the drug, as such events may also be caused by the natural progression of the disease. In addition, we were unable to examine the association with comorbidities that could cause gastrointestinal bleeding, such as the presence of arteriosclerosis and *Helicobacter pylori* infection, because JADER is a database where information on adverse drug reaction cases is reported voluntarily, and, as a result, we could not obtain sufficient information. Second, cases in the JADER database are spontaneously reported, hence, reporting bias cannot be ruled out. Furthermore, the classification of TKI-induced gastrointestinal bleeding events was not reported in the JADER database, and taking specific measures against TKI-induced gastrointestinal bleeding was challenging. Only data on patients with side effects are available, and there is concern that research data may target a patient group distinct from those often present in actual clinical practice. Therefore, the results of this study need to be further expanded through cohort studies and randomized controlled trials. Furthermore, the frequency of reported gastrointestinal bleeding events with bosutinib was similar to that of imatinib, and it might be possible that signals associated with gastrointestinal bleeding were not detected due to fewer reports on bosutinib. In addition, we were unable to detect the effects of concomitants with medications related to bleeding due to the small number of studies. Therefore, it is necessary to perform further analyses after accumulating more data.

## 5. Conclusions

Gastrointestinal bleeding events associated with TKIs were investigated using the JADER database. We found that specific TKIs and patient characteristics were associated with gastrointestinal bleeding. In particular, the association of gastrointestinal bleeding with dasatinib was affected by sex (female) and age (≥60 years). These patient characteristics should be considered to avoid the onset of gastrointestinal bleeding when prescribing TKIs.

## Figures and Tables

**Figure 1 medicina-58-01495-f001:**
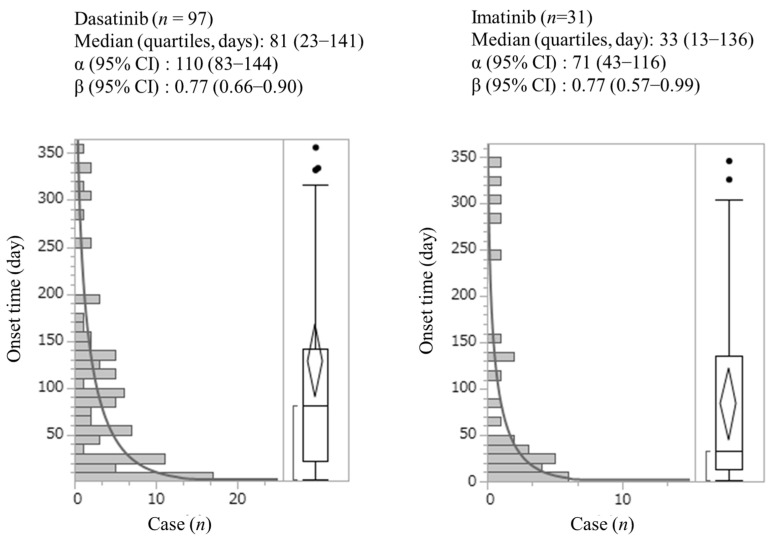
Number of gastrointestinal bleeding cases associated with dasatinib and imatinib by onset time in the JADER database. The time-to-onset dasatinib was 81.0 (23.0–141.0) days. The time-to-onset imatinib 33.0 (13.0–136.0) days. Black dots showed outliers.

**Figure 2 medicina-58-01495-f002:**
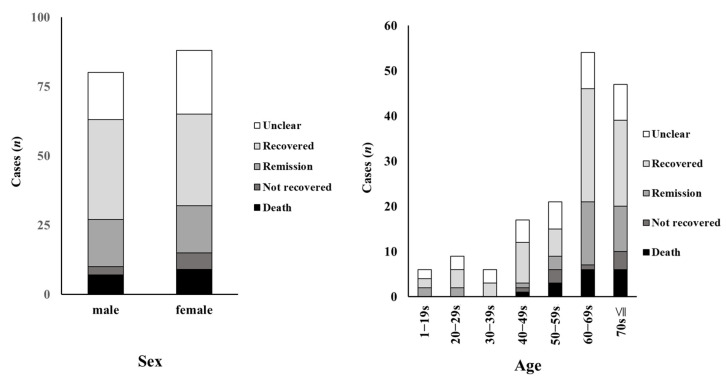
Analysis of age and outcomes in cases with dasatinib-associated gastrointestinal hemorrhage. The outcome profiles of cases with dasatinib-associated gastrointestinal bleeding are summarized according to sex, age, and outcome.

**Table 1 medicina-58-01495-t001:** Reported cases and the crude reporting odds ratios (RORs) of gastrointestinal bleeding.

		Total	Cases	Non-Cases	Ratio (%)	Crude ROR (95% CI)	*p*-Value
Total		595,121	15,418	579,703	2.59		
Dasatinib	Total	1507	159	1348	10.55	4.47 (3.77–5.28)	<0.001
	0–59 years	646	58	588	8.98	6.30 (4.72–8.28)	<0.001
	≥60 years	861	101	760	11.73	4.07 (3.27–5.02)	<0.001
	Male	873	81	792	9.28	3.43 (2.70–4.33)	<0.001
	Female	634	78	556	12.30	6.15 (4.78–7.81)	<0.001
Imatinib	Total	3896	122	3774	3.13	1.22 (1.01–1.46)	0.038
	0–59 years	1367	35	1332	2.56	1.66 (1.15–2.33)	0.0058
	≥60 years	2529	87	2442	3.44	1.09 (0.87–1.35)	0.46
	Male	2229	68	2161	3.05	1.05 (0.81–1.34)	0.66
	Female	1667	54	1613	3.23	1.46 (1.09–1.91)	0.010
Bosutinib	Total	334	12	322	3.59	1.40 (0.72–2.48)	0.23
	0–59 years	112	1	111	0.89	0.57 (0.014–3.23)	0.99
	≥60 years	222	11	211	9.91	1.59 (0.78–2.90)	0.13
	Male	216	8	208	3.70	1.28 (0.55–2.58)	0.42
	Female	118	4	114	3.39	1.52 (0.41–4.01)	0.34
Nilotinib	Total	1587	27	1560	1.70	0.65 (0.43–0.95)	0.026
	0–59 years	511	6	505	1.17	0.75 (0.27–1.64)	0.59
	≥60 years	1076	21	1055	1.95	0.61 (0.37–0.93)	0.019
	Male	988	17	971	1.72	0.58 (0.34–0.94)	0.023
	Female	599	10	589	1.67	0.74 (0.35–1.36)	0.41

ROR: reporting odds ratio; 95% CI: 95% confidence interval. Categorical variables were analyzed using Fisher’s exact test. Statistical significance was defined as *p <* 0.05. Cases: cases with reported gastrointestinal bleeding; Non-cases: cases with reported adverse effects other than gastrointestinal bleeding.

**Table 2 medicina-58-01495-t002:** Multiple logistic regression analysis.

	Adjusted ROR (95% CI)	*p*-Value
Total		
Reporting year	1.04 (1.04–1.05)	<0.001
Age ≥60 years, *n* (%)	2.22 (2.1–2.36)	<0.001
Male sex, *n* (%)	1.47 (1.37–1.57)	<0.001
Dasatinib	8.02 (5.75–10.20)	<0.001
Imatinib	1.81 (1.2–2.72)	0.0045
Nilotinib	0.57 (0.21–1.51)	0.26
Bosutinib	0.38 (0.046–3.13)	0.37
Age ≥60 years and Male sex	0.82 (0.76–0.89)	<0.001
Age ≥60 years and Dasatinib	0.69 (0.49–0.97)	0.035
Age ≥60 years and Imatinib	0.70 (0.47–1.05)	0.083
Age ≥60 years and Nilotinib	0.97 (0.38–2.43)	0.94
Age ≥60 years and Bosutinib	3.70 (0.47–29.50)	0.22
Male sex and Dasatinib	0.57 (0.41–0.8)	<0.001
Male sex and Imatinib	0.76 (0.52–1.09)	0.14
Male sex and Nilotinib	0.93 (0.42–2.06)	0.86
Male sex and Bosutinib	0.86 (0.25–3.0)	0.82

ROR: reporting odds ratio; 95% CI: 95% confidence interval. Categorical variables were analyzed using Fisher’s exact test. The RORs and 95% CIs were calculated for each explanatory variable in the multivariate model. Statistical significance was defined as *p <* 0.05.

## Data Availability

The data presented in this study are available on request from the corresponding author.

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
