# Peer review of "Adverse Reaction Profiles Related to Gastrointestinal Bleeding Events Associated with BCR-ABL Tyrosine Kinase Inhibitors"

_medicina, 2022, doi:10.3390/medicina58101495_

Round 1

Reviewer 1 Report

Thanks for the opportunity to review this interesting study looking at reported GI bleedings in response to TKI therapy in a large Japanese pharmacovigilance database. The authors have asked an interesting question of a large dataset, and obtained useful data that adds to our knowledge in the area. However, some review of the statistical analysis and interpretation, and the resulting discussion is needed before the paper can be published.

1. The conclusion that TKI-associated  bleeding is more common in men is erroneous. All the ROR are <1.0, both on univariate and multivariate analysis, and the % is lower for both dasatinib and imatinib. This conclusion needs to be changed, and the analysis reconsidered (eg page 7 line 232 add "and in females", page 8 line 273 and following - rewrite to correct.) 

2. There was a similar ROR of bleeding with bosutinib to imatinib in the study - only the number of bleeding cases reported with bositinib did not reach statistical significance. The correct interpretation of this data is that there is currently no evidence from this study that bosutinib is safer than imatinib, but more data are needed. This should be added to discussion.

3. The method of multivariate analysis appears some what idiosyncratic, as factors such as sex and age have been considered as variables of interest rather than covariates. This might be worth discussing with a statistician.

4. Several times reference is made to bleeding being "more severe" (page 7 line 231, page 8 line 283)but I cannot see any evidence that severity of bleeding was measured in the study or analysed. Perhaps the authors mean "more frequent"?

Thanks again for the opportunity to review this study.

Reviewer 2 Report

The article entitled “Adverse reaction profiles related to gastrointestinal bleeding events associated with BCR-ABL tyrosine kinase inhibitors” by Kei Kawada, et al. demonstrated that Specific TKIs and patient characteristics were associated with gastrointestinal bleeding.

This study has some value, however, the discussion about the background of patients is not enough. As a result, the authors present a limited interest.

Thus, there are areas that need to be improved.

Major comments

1.     The authors should provide the mention about the background in patients treated with TKIs. Please discuss about the relationship between medications (e.g., antiplatelet agents, anticoagulant agents, protompomp inhibitors) other than tyrosine kinase inhibitors and gastrointestinal bleeding rate. In this study, there is not enough mention about comorbidities that can cause gastrointestinal bleeding, such as arteriosclerosis, Helicobacter pylori infection.

2.     It is unclear why findings of this study can be applied to minimize the risk of gastrointestinal bleeding in patients treated with TKIs. Therefore, gastrointestinal bleeding events with TKIs should be classified (e.g., ulcers, telangiectasias, gastrointestinal malignancy).

3.     Please discuss about comparison of gastrointestinal bleeding rates with the previous reports. It is not clear how the conditions and results differ from them.

Minor comments

1.     In line 25-26, “Place the question addressed in a broad context.” is misprint.

2.     Descriptions in the “Materials and Methods” ( Line 67-172 ) are somewhat redundant and need shortening.

Round 2

Reviewer 1 Report

The authors have satisfactorily addressed my concerns.

Author Response

Thank you for your helpful comments.

Reviewer 2 Report

The author's response to reviewers' comments was appropriate, and the revised manuscript was appropriately refined. However, the author must respond to the next minor comment. As a result, the report is of great concern.

Minor comments

Is the text in lines 1.320-322 correct?

Consider that the author's response to the previous reviewer's main comment (2), sentences in lines 343-344, should be replaced with other content.

(previous main comments "2"; It is unclear why the findings of this study can be applied to minimize the risk of gastrointestinal bleeding in patients treated with TKI. Therefore, gastrointestinal bleeding events with TKIs (e.g., ulcers, telangiectasia, gastrointestinal malignancies) should be classified.

Author Response

Thank you for your helpful comments. We have thoroughly reviewed the entire manuscript and carefully responded to all comments. Our point-by-point responses are listed below.

Comments and Suggestions for Authors

The author's response to reviewers' comments was appropriate, and the revised manuscript was appropriately refined. However, the author must respond to the next minor comment. As a result, the report is of great concern.

Minor comments

Is the text in lines 1.320-322 correct?

Response:

Thank you for your comment. Since this statement was a prediction and lacked accuracy, we removed it from the revised manuscript.

Discussion (page 9, lines 267–269 in the revised version of the manuscript)

The type of TKIs used may have varied depending on the patient's condition, and these factors may also have affected TKI-induced gastrointestinal bleeding.

Consider that the author's response to the previous reviewer's main comment (2), sentences in lines 343-344, should be replaced with other content.

Response:

Thank you for your valuable comment. As suggested, we have changed the sentences in lines 343-344 (revised manuscript: lines 338-341) as follows:

(Main comment 2 in the previous letter: “It is unclear why the findings of this study can be applied to minimize the risk of gastrointestinal bleeding in patients treated with TKI. Therefore, gastrointestinal bleeding events with TKIs (e.g., ulcers, telangiectasia, gastrointestinal malignancies) should be classified.”)

Conclusions (page 10, lines 290–291)

Our findings can be applied to minimize the risk of gastrointestinal bleeding in patients treated with TKIs.

These patient characteristics should be considered to avoid the onset of gastrointestinal bleeding when prescribing TKIs.
